# Silage Mixtures of Alfalfa with Sweet Sorghum Alter Blood and Rumen Physiological Status and Rumen Microbiota of Karakul Lambs

**DOI:** 10.3390/ani12192591

**Published:** 2022-09-28

**Authors:** Jiao Wang, Long Cheng, Abdul Shakoor Chaudhry, Hassan Khanaki, Imtiaz H. R. Abbasi, Yi Ma, Farzana Abbasi, Xuefeng Guo, Sujiang Zhang

**Affiliations:** 1Key Laboratory of Tarim Animal Husbandry Science and Technology, College of Animal Science and Technology, Tarim University, Alar 843300, China; 2Faculty of Veterinary and Agricultural Sciences, The University of Melbourne, Dookie College 3647, Australia; 3School of Natural and Environmental Sciences, Newcastle University, Newcastle upon Tyne NE1 7RU, UK; 4Department of Animal Nutrition, Faculty of Animal Production and Technology, Cholistan University of Veterinary and Animal Sciences, Bahawalpur 63100, Pakistan; 5Laboratory of Metabolic Manipulation of Herbivorous Animal Nutrition, College of Animal Science and Technology, Yangzhou University, Yangzhou 225009, China; 6Faculty of Chemical and Biological Sciences, The Islamia University, Bahawalpur 63100, Pakistan

**Keywords:** silage mixture, alfalfa, sweet sorghum, rumen fermentation, rumen microbiota, Karakul lambs

## Abstract

**Simple Summary:**

The study aimed to detect the effects of alfalfa–sweet sorghum mixed silages on blood and rumen physiological status, and rumen microbiota of sheep. The results demonstrated that 60% inclusion of alfalfa in the alfalfa–sweet sorghum mixed silage appeared to be the best in improving serum antioxidant capacity, dry matter intake, rumen fermentation and cellulolytic bacteria abundance of Karakul lambs. The result of this study is useful for understanding and promoting the utilization of suitable mixtures of alfalfa–sweet sorghum silage in animal production.

**Abstract:**

The study investigated the effects of feeding mixtures of alfalfa (AF) and sweet sorghum (SS) at different ratios of silages in terms of the physiological status of blood and rumen, and rumen microbiota in lambs. A total of 30 four-month-old male Karakul lambs with 25.5 ± 1.4 kg mean initial body weight were randomly allocated to five groups, with six lambs in each group. Five experimental diets containing 40% of one of the five AF–SS mixed silages (containing 0%, 20%, 40%, 60%, and 80% AF on a fresh weight basis, respectively) and 60% of other ingredients were formulated. Overall, the results showed that the mixed silage with more AF tended to increase serum antioxidant capacity, dry matter (DM) intake, and rumen fermentation metabolites. The AF–SS mixed silages containing AF at 60% and 80% caused a significant linear increase (*p* < 0.05) in the activity of total antioxidant capacity. The superoxide dismutase in the Karakul lamb responded with significant linear and quadratic increases (*p* < 0.01) as the ratio of AF was increased in the AF–SS mixed silages. Feeding diets with AF in silage mixtures at the ratio of 60% significantly increased (*p* < 0.05) the concentration of ruminal total volatile fatty acids (tVFA), acetate, and ammonia-N. However, no statistical significance (*p* > 0.05) was found in the alpha diversity of rumen microbes among the tested groups (*p* > 0.05). Principal coordinates analysis could clearly discriminate the differences between the five groups (*p* = 0.001). The relative abundance of *Firmicutes* in the rumen were significantly higher with AF at 40% in the AF–SS silage-based diet than those with AF at 0%, and 20% ratios. The abundance of *Ruminococcus_albus* had a significant linear increase (*p* < 0.05), as the ratio of AF in the AF–SS mixed silages was increased. In conclusion, the best beneficial effect on the physiological status of the blood and rumen, DM intake, and rumen microbiota in lambs came from those that consumed the diet containing the AF–SS mixed silage with 60% AF.

## 1. Introduction

Rapid development of ruminant husbandry in many developing countries and increasing prices of forages due to their shortage in arid areas have become keys factor in restricting the sustainable development of the ruminant industry [1,2]. One option to increase forage production in arid areas for ruminants is to cultivate more water-use-efficient crops such as sweet sorghum (*Sorghum bicolor*; SS) [3].

As a promising forage to be used as a suitable feed for livestock production in arid regions, SS has attracted more and more attention in recent years because it has not only higher tolerance to salty, alkaline and drought conditions [4,5], but also high water-use efficiency, high biomass yield and high sugar content [6,7]. It is a suitable material for making silage due to its rich sugar content and high biomass, but it is difficult for SS silage alone to provide sufficient crude protein (CP) for animal growth and production [8,9]. Alfalfa (*Medicago sativa*; AF) is a perennial leguminous forage with rich protein content. Although making AF silage is good for preventing nutrient loss and increasing palatability, it is not easy to successfully conserve because it has low sugar, high protein, high buffering capacity and high protein decomposition in the silage-making process [10,11].

Combining AF with SS to make AF–SS mixed silages can improve the nutritional quality and fermentable characteristics of the mixed silages, which not only increases the success rate of AF silage, but also make up the deficiency of insufficient protein in SS silage alone. A study [8] showed that degradability of dry matter (DM), organic matter and total volatile fatty acids (tVFA) in vitro were affected by the percentage of AF in the AF–SS mixed silages. The use of AF–SS silage mixtures as animal feedstuff has also been studied in sheep [9]. This research indicated that the ratio of AF to SS in the mixed silages could affect not only the quality of AF–SS silage mixtures, but also the growth of sheep. It is widely believed that dietary energy and protein levels can affect production traits via changing rumen microbial populations [12,13]. In addition, blood physiological status is an important indicator of animal health [14]. However, little research is known about the effects of feeding AF–SS mixed silages with different ratios of AF to SS in terms of the blood and rumen physiological status, and rumen microbiota. Therefore, the purpose of this study was to explore the influences of AF–SS mixed silages on the physiological status of blood and rumen, and rumen microbiota of sheep.

## 2. Materials and Methods

### 2.1. Silage Mixtures and Experimental Diets

The varieties of AF (*Hetian Big-leaf*) and SS (*Cowley*) were cultivated and harvested at Alar, XinJiang, China. The whole fresh plant of AF and SS were harvested and chopped to 2.5 cm using a multi-function chopper. According to the different ratios of AF to SS (0%AF:100%SS, 20%AF:80%SS, 40%AF:60%SS, 60%AF:40%SS, and 80%AF:20%SS based on fresh weight), five AF–SS silage mixtures were made using an ensiling machine (Qufu Tianliang Trading Co., Ltd., Shandong, China) [8]. The silage mixtures were stored in polythene bags at room temperature for 60 days until use. Five lamb diets containing 40% of the mixed silages, representing different AF–SS ratios (based on DM), were blended with 60% of other ingredients by using a ration mixer (RH-IBJ-400, Qufu Runhua Machinery Manufacturing Co., Ltd., Shandong, China). The ingredients and chemical compositions of the five experimental diets are presented in Table 1.

### 2.2. Animal Feeding and Sampling

The present study was part of an animal experiment conducted at the Animal Research Station, Tarim University. All procedures and animal care were performed in compliance with the Guidelines for the Care and Use of Animals for Research in China (GB 14925-2001). This experimental protocol (No. 2019-25) used in this study was approved by the Animal Research Ethics Committee of Tarim University. Assignment to groups, management, housing conditions, and slaughter are described in detail in Wang et al. [9], where growth trait, nutrient digestibility, and meat quality were reported. Briefly, thirty (4-month-old) healthy Karakul lambs with similar initial body weight (25.5 ± 1.4 kg) were divided into five groups of six lambs per group. The lambs were housed in individual stalls equipped with water buckets and feeders. The lambs were allowed ad libitum access to the respective diets and water in their allocated stalls. After every 24 h, the amount of each refused feed was collected, weighed, recorded and replaced with fresh portions of respective experimental diets.

After 80 days of experiment, blood samples (10 mL) from the jugular vein of each lamb were collected at 08:00 and centrifuged at 3000× *g* for 10 min at 4 °C to obtain serum. Serum samples were frozen at −20 °C until analysis. All lambs were slaughtered at the end of the study. About 20 mL rumen fluid post mortem was sampled from the ventral part of the rumen. The rumen fluid was collected after filtration through four layers of gauze, and the pH was measured immediately using an acidity meter (PHS-100, Tianqi Mdt Info Tech, Ltd., ShangHai, China). Subsamples of separate rumen fluid were frozen on dry ice and transported to the lab where they were stored at −80 °C until DNA extraction and subsequent analysis.

### 2.3. Analysis of Blood Physiological Status

The concentrations of creatinine, total cholesterol (T-CHO), blood urea nitrogen (BUN), the activities of total antioxidant capacity (T-AOC) and superoxide dismutase (SOD) in serum were determined by commercial kits (Jiancheng Biological Technology Co., Ltd., Nanjing, China) by following the manufacturer’s instructions [15].

### 2.4. Analysis of Rumen Physiological Status

The ammonia–nitrogen compound (NH_3_-N) was analyzed by the phenol-hypochlorite reaction according to the steps outlined by Weatherburn [16]. Rumen liquid (10 mL) was mixed with 2.5 mL of a HPO_3_ (250 g/L), and then centrifuged at 4 °C for 10 min (12,000× *g*) for tVFA analysis. Rumen tVFA (i.e., acetate, propionate, butyrate, valerate, isobutyrate, and isovalerate) analysis was performed by gas chromatography (GC-2014FRGA1, Shimadzu, Tokyo, Japan). The injector temperature was 250 °C, the oven and the detector were maintained at 160 °C. The split injection was 1:25 whereas nitrogen as the carrier gas (flowrate 0.6 mL/min) and make-up gas (flowrate 25 mL/min) were used. The flowrates of hydrogen and air were 20 and 300 mL/min, respectively [17].

### 2.5. DNA Extraction, 16S rDNA Gene Amplification, Sequence Processing and Analysis

The DNA samples of ruminal microbial solution were directly extracted with TIANamp Stool DNA Kit (TIANGEN, Beijing, China). The 16S rDNA V3-V4 regions were amplified by PCR using primers 338F 5′-ACTCCTACGGGAGGCAGCAG-3′ and 806R 5′-GGACTACHVGGGTWTCTAAT-3′ (synthesized by Biological Engineering Co., Ltd., Shanghai, China) [18]. The total volume of the reaction mixture was 50 µL, which consisted of 0.35 µg of template DNA, 2 µL primer mix (10 µM), 5µL 10 × Taq Buffer (TIANGEN, Beijing, China), 4 µL dNTP mixture (2.5 mM), 0.5 µL DNA polymerase (2.5 U/µL, TIANGEN, Beijing, China), and approximately 38.5 µL milli-Q water. Thermocycling parameters were as follows: initiated at 95 °C for 5 min to denature, followed by 35 cycles at 95 °C for 30 s, 50 °C for 30 s and 72 °C for 90 s, and extended at 72 °C for 10 min. Finally, 3 amplicons were selected for each group and delivered to Novogene Technology Co. (Beijing, China), then sequenced on an Illumina Platform according to the standard procedures.

Raw reads were filtered to obtain the final clean reads using FASTP [19], then the noisy sequences of raw tags were analyzed using QIIME (version V1.9.1) [20]. After removal of chimeric sequences with VSEARCH [21], all clean reads were clustered into operational taxonomic units (OTUs) using Uparse software [22] at a similarity of 97%. The most abundant sequences in OTUs were screened out as the representative sequences by the Silva database 132 [23], according to the reference taxonomy provided by the SSU rRNA database. Five alpha indices including Observed species, Shannon, Simpson, Chao1, and ace were calculated. Principal coordinates analysis (PCoA) based on Bray–Curtis distance was estimated to reveal beta diversity in bacterial communities among the five groups. The bacterial compositions of each group at phylum and genus level were conducted using OriginPro software (version 9.0) to obtain relative abundance histograms.

### 2.6. Statistical Analysis

All the data were analysed using SPSS software (version 26.0; SPSS Inc., Chicago, IL, USA). One-way ANOVA was used to examine the linear and quadratic effects of silage types on the physiological status in blood and rumen, and rumen microorganisms. The Duncan’s post hoc test was used for multiple comparisons of means. Group differences were declared significant at *p* < 0.05.

## 3. Results

### 3.1. Physiological Status in Blood

The effects of AF–SS silage-based diets on serum biochemical parameters of Karakul lambs are given in Table 2. The contents of creatinine, BUN and T-CHO were not affected significantly (*p* > 0.05) by diets, whereas the activities of T-AOC and SOD differed significantly among diets (*p* < 0.05 and *p* < 0.01; respectively). Animals fed on AF–SS silage-based diets with AF at 60% and 80% inclusion had a significantly (*p* < 0.05) higher activity of T-AOC compared to diets with AF at 0%, and 20% ratio. The SOD activity of lambs fed diets with AF at 40% and 60% was significantly (*p* < 0.01) higher than that of those with AF at 0%, 20% and 80%.

### 3.2. DM Intake and Physiological Status in Rumen

The effects of AF–SS silage-based diets on DM intake, ruminal pH, NH_3_-N and tVFA concentrations of Karakul lambs are presented in Table 3. The DM intake of lambs was significantly (*p* < 0.01) higher with AF at 60% and 80% in the AF–SS silage mixtures than those with AF at 0%, 20% and 40% inclusions. The ruminal NH_3_-N concentration of Karakul lambs fed diets containing mixed silages with AF at 40%, 60% and 80% were significantly (*p* < 0.05) higher than those with AF at 0% and 20% inclusions. In general, the ruminal tVFA and acetate of the animals showed a significant (*p* < 0.05) linear and quadratic increase as the ratio of AF increased in the AF–SS mixed silages, and the values were significantly (*p* < 0.05 and *p* < 0.01; respectively) higher with a ratio of AF at 60% in the AF–SS mixed silages than those with AF at 0% and 20% ratios. However, there were no significant (*p* > 0.05) differences in the ruminal concentrations of tVFA and acetate between diets with AF at 40% and those at 60%. For the items of ruminal pH, propionate, butyrate, valerate, isobutyrate, isovalerate, and ratio of acetate to propionate, the difference was not significant (*p* > 0.05).

### 3.3. Alpha Diversity and Analysis of OTUs

The effects of AF–SS silage-based diets on the alpha diversity of Karakul lambs are presented in Table 4. No statistically significant (*p* > 0.05) difference was found in alpha diversity of rumen microbes among groups. The index of Shannon numerically tended to increase for lambs fed AF–SS silage-based diets with higher ratio of AF in the mixed silages.

The diagrams were formed by analyzing common and unique OTUs among the five groups. As shown in Figure 1a, there were 2614, 2405, 2153, 2698, and 2373 OTUs in each group, with unique OTUs of 224, 166, 130, 286 and 168, respectively. The PCoA showed that the ruminal microbiota from the five treatments were separated based on Bray–Curtis similarity method (*p* = 0.001, Figure 1b).

### 3.4. Ruminal Bacterial Communities at Phylum

In the rumen, 28 phyla were found. Data from top 10 ruminal microbiota were analyzed (Figure 2) and the results showed that *Bacteroidetes* (42.44–59.95%), and *Firmicutes* (34.82–51.66%) had relatively higher abundances.

As shown in Table 5, the abundance of *Firmicutes* in the rumen with a percentage of AF at 40% in the AF–SS silage-based diet was significantly (*p* < 0.05) higher than those with AF at 0%, and 20% ratios. The relative abundance of *Actinobacteria* and *Melainabacteria* in the rumen responded with a significant (*p* < 0.05) linear increase for lambs fed diets containing 60% and 80% AF. The ruminal relative abundances of *unidentified_Bacteria* and *Euryarchaeota* from lambs fed AF–SS silage-based diets with 40% AF in the mixed silages were significantly (*p* < 0.05) greater than those with 0%, 20% and 80% AF inclusion.

### 3.5. Ruminal Bacterial Communities at Genus and Species

About 441 genera were found, and the top 20 genera with high abundances are displayed in Figure 3. The results showed that *unidentified_Bacteroidales* (3.30–21.51%) was the main abundant genus in the rumen, followed by *unidentified_Lachnospiraceae* (2.26–6.49%).

When related to the abundance of ruminal bacterial communities at genus level (Table 6), the relative abundances of *unidentified_Bacteroidales* from lambs fed the AF–SS silage-based diets with 0% and 20% AF were linearly (*p* < 0.05) higher than those with 40%, 60% and 80% AF in the mixed silages. The highest ruminal relative abundance of *Succiniclasticum* in lambs fed AF–SS silage-based diets came from those fed diets containing 40% AF in the mixed silage, followed by 60% AF and 20% AF, which were significantly higher than those from those with 0% AF and 80% AF (*p* < 0.05). The relative abundance of *Acetitomaculum* and *Enterorhabdus* had a significant (*p* < 0.01) linear increase for lambs fed AF–SS silage-based diets containing 60% and 80% AF.

The relative abundances of rumen bacterial species are presented in Table 7. The diet including the mixed silage with 80% AF was promoted for the higher relative abundance of *Ruminococcus_albus* compared with those with 0% AF (*p* < 0.05). There were no significant differences among the abundances of *Ruminococcus_flavefaciens*, *Butyrivibrio_fibrisolvens*, and *Clostridium_disporicum* (*p* > 0.05).

## 4. Discussion

### 4.1. Physiological Status in Blood

Oxidative stress can inhibit the growth of ruminants and lead to production loss [24]. The antioxidant enzymes can eliminate free radicals to increase the antioxidative function of animals [25]. In the present study, the activity of T-AOC was higher in lambs fed with AF at 60% and 80% inclusion, and the activity of SOD was higher with AF at 40% and 60% inclusion. The higher values of T-AOC and SOD of lambs fed diets containing mixed silages with more AF, indicating better antioxidant status, might be due to the higher concentration of saponins and flavonoids in the mixed silage with more AF. The AF may be a source of natural antioxidants to relieve oxidative stress in animal feeding, which WAS associated with the increased SOD expression [26]. Chen et al. [27] reported that flavonoids extracted from AF exhibited powerful antioxidant activities. Zhang et al. [8] also found that saponins content was improved with the increased ratio of AF in the AF–SS mixed silages.

### 4.2. DM Intake and Physiological Status in Rumen

The higher levels of DM intake and ruminal NH_3_-N concentration of lambs fed the AF–SS silage-based diets with high AF ratio in the silage mixtures were expected. It was likely to be a result of being fed higher levels of dietary CP, which had a positive correlation with DM intake and rumen NH_3_-N concentration [28]. These results reflected that a greater proteolysis occurred during rumen fermentation of lambs fed AF–SS silage-based diets, as mixed silages contained a greater AF ratio. This is in accordance with Tao et al. [29], who observed that the ruminal NH_3_-N content was decreased linearly as corn increased and AF decreased in steers. A decreased rumen NH_3_-N concentration was also observed in Holstein cows supplemented with dry sugar beet pulp [30] and heifers supplemented with fructose [31].

The ruminal tVFA and acetate were increased with the increase of AF in mixed silage, which probably were due to the greater CP content, creating a more favorable environment in the rumen for the growth of cellulolytic bacteria. Polyorach et al. [32] have showed that the higher ruminal ammonia caused the increased cellulolytic bacterial population. Ammonia in rumen can provide a nitrogen source for rumen ecology [33]. There was a positive correlation between the cellulolytic bacteria, acetate and propionate in rumen [34]. In this study, when lambs were fed silage-mixed diets with higher AF, rumen tVFA and acetate concentrations were increased, which may be due to the increased number of cellulolytic bacteria. Besides that, the increase of minerals concentrations in mixed silages with higher AF [8] may be beneficial to the growth of cellulolytic bacteria [35]. Similar to the results of the current study, Benchaar et al. [36] reported that replacing corn silage with AF silage increased the ruminal acetate in dairy cows. Branched-chain VFA were derived from the degradation of branched-chain amino acids [37]. Niyigena et al. [38] found that the concentration of ruminal tVFA was linearly decreased, with increasing fescue concentration in the tall fescue–alfalfa silage mixtures. Consistent with our findings, another study [39] observed that AF could increase the concentration of ruminal tVFA. Additionally, the concentration of NDF in high-AF-based diets was much lower than that in high-SS-based diets, which had a negative relationship with VFA generation [40], resulting in an increase in tVFA in the higher AF-based diets.

### 4.3. Ruminal Bacterial Communities

The results showed that the AF–SS mixed silages had no impact on alpha diversity of the rumen microbiota, indicating that the community diversity and community richness were not affected by the ratio of AF to SS in mixed silages. The PCoA analysis showed that the tested rumen bacterial community could be grouped into five clusters according to the Bray–Curtis distance metric. It indicated that feeding different ratios of AF in the AF–SS silage-based diets had an effect on the microbiota composition of lambs. Du et al. [41] reported that the composition of the bacterial community of lambs fed diets with 10%, 20%, 30% AF inclusion was distinctly separated according to PCoA analysis, which indicated that different bacteria prefer different nutrients and ruminal environments.

Regardless of diet, the 16S rDNA high-throughput sequencing technology revealed that the microbial community of the lamb was dominated by *Bacteroidetes* and *Firmicutes* at the phylum level (Figure 2). Similar findings have been observed in Karakul sheep [42] and other sheep [43,44]. For the ruminants, *Firmicutes* play a considerable role in degradation of the fiber and cellulose [45]. The higher relative abundances of *Firmicutes* in the higher ratio of AF (40%, 60% and 80%) with AF–SS silage-based diets could promote fibre digestion because of their dominance in the ruminal microbiota. In accordance with a previous study [9], higher AF based diets reflected higher neutral-detergent fibre (NDF) and acid-detergent fibre (ADF) digestibility, as well as the higher DM intake. Belanche et al. [46] reported that cellulolytic bacteria (i.e., *Ruminococcus_albus*, *Ruminococcus_flavefaciens*, *Firbrobacter_succinogenes*, and *Butyrivibrio_fibrisolvens*) seem to be particularly sensitive to the shortage of nitrogen, and microbial diversity was lower in the low-protein diets compared to the high-protein diets. The higher relative abundance of *Ruminococcus_albus* with AF at 80% inclusion may be due to the higher CP contents in AF than that in SS, which was reported by Zhang et al. [8]. Similarly, da Silva-Marques et al. [47] also demonstrated that high protein supplement provided higher ruminal NH_3_ and proportion of *Ruminococcus_albus* for the growth of cellulolytic bacteria of bulls when compared with medium protein supplement. Another study also found that an increase in dietary energy levels promoted rumen energy productivity and microbial protein yield [48]. The results are consistent with our research.

*Succiniclasticum* were starch-degrading bacteria that can produce acetate and succinic acid during starch degradation [49,50]. Ruminal succinic acid was eventually transformed into propionate to provide energy for microbial protein synthesis [51]. In this study, the greater ruminal relative abundance of *Succiniclasticum* came from including 20% AF, 40% AF and 60% AF in the mixed silages. This indicated that the content of starch in the AF–SS silage-based diets with AF at 20%, 40% and 60% in the mixed silages was more suitable for the growth of *Succiniclasticum* in the rumen of lambs. Furthermore, no benefit from more than 60% AF in the mixed silages for further decreasing fibre content (NDF and ADF) was observed (Table 1), which might be another reason for the less relative abundance of Succiniclasticum from diet with mixed silage containing 80% AF. Meanwhile, the polyphenols from plant-based diets could inhibit rumen fermentation by interfering with the bacterial cell wall or affecting ruminal microbe enzymes [52]. Therefore, lower contents of ruminal acetate and tVFA from the AF–SS silage-based diets with a larger SS ratio would be reasonable because polyphenol-rich SS may depress rumen fermentation.

Notably, a large number of unidentified and uncultured genera in the rumen of lambs were detected in this study, indicating that the lambs might have a more diverse ruminal microbiome, but that only a little part of the microbe has been detected or identified. This is a clear gap that we need to search more in the future studies.

## 5. Conclusions

This study showed that the physiological status of blood and rumen, and rumen microbiota of Karakul lambs can be modified by feeding diets containing silages of different AF and SS ratios. The results demonstrated that an increase in AF ratio in the AF–SS mixed silages appeared to generally improve serum antioxidant capacity, DM intake, ruminal tVFA, acetate and cellulolytic bacteria in Karakul lambs. Overall considering the results, the inclusion rate of 60% AF is recommended as the best amount in the AF–SS mixed silages for optimal physiological status in blood and rumen, and rumen microbiota. Further studies are suggested to explore the best dietary supplements to enhance the effect of similar mixed silages on the health and production of ruminant animals.

## Figures and Tables

**Figure 1 animals-12-02591-f001:**
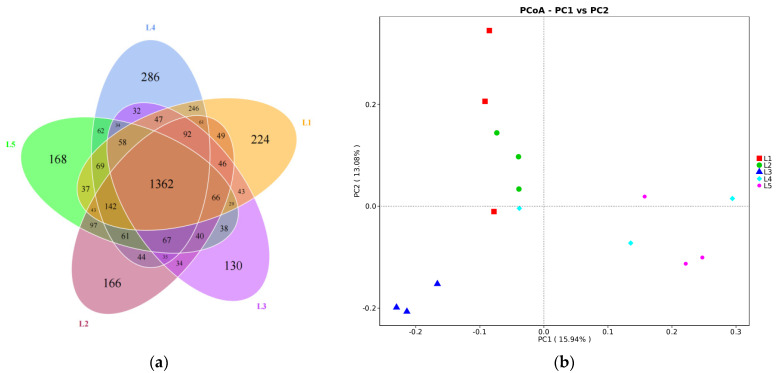
Petal diagrams (**a**) of OTUs for the rumen of the lamb. PCoA (**b**) plot based on the Bray–Curtis dissimilarity of the microbiota (*p* = 0.001). L1, L2, L3, L4, and L5 represent five groups of lambs treated with dietary alfalfa (AF) ratio at 0%, 20%, 40%, 60%, and 80% in the alfalfa–sweet sorghum (AF–SS) silage mixtures; n = 3 for each group.

**Figure 2 animals-12-02591-f002:**
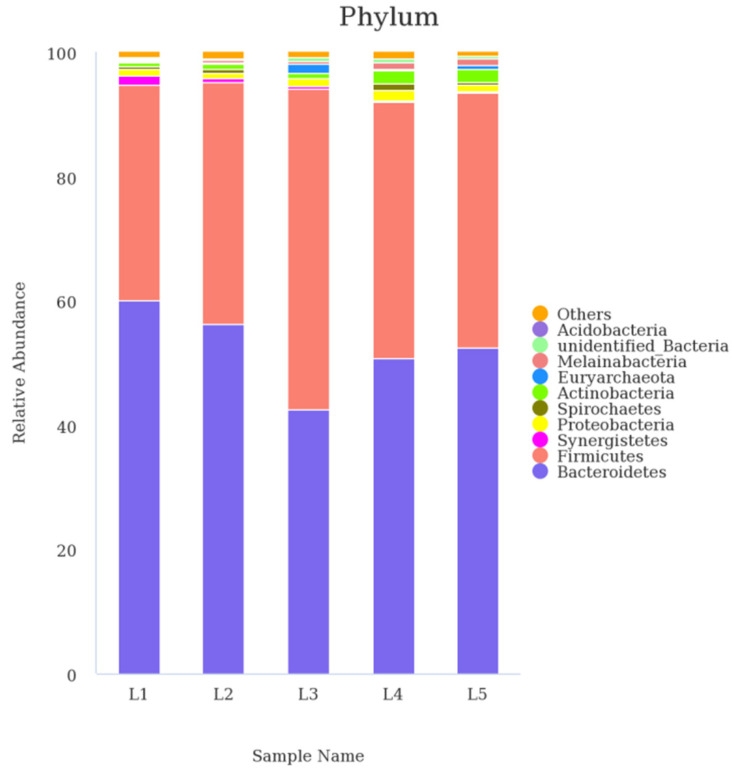
The column chart of the main dominant phylum (only the top 10 abundant phyla are presented). L1, L2, L3, L4, and L5 represent five groups of lambs treated with dietary alfalfa (AF) ratio at 0%, 20%, 40%, 60%, and 80% in the alfalfa–sweet sorghum (AF–SS) silage mixtures; n = 3 for each group.

**Figure 3 animals-12-02591-f003:**
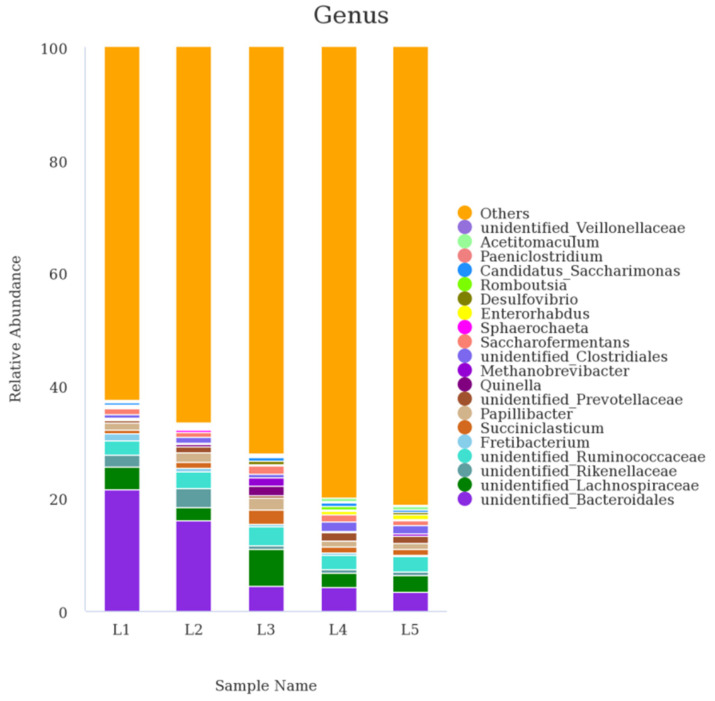
The column chart of the main dominant genus (only the top 20 abundant genera are presented). L1, L2, L3, L4, and L5 represent five groups of lambs treated with dietary alfalfa (AF) ratio at 0%, 20%, 40%, 60%, and 80% in the alfalfa–sweet sorghum (AF–SS) silage mixtures; n = 3 for each group.

**Table 1 animals-12-02591-t001:** Ingredients and chemical compositions of alfalfa–sweet sorghum (AF–SS) silage-based diets.

Items	AF Percentage in the Silage Mixtures				
	0%AF	20%AF	40%AF	60%AF	80%AF
Ingredients					
Silage mixtures	40	40	40	40	40
Rice straw	10	10	10	10	10
Cottonseed hull	10	10	10	10	10
Corn	13	13	13	13	13
Soybean meal	12	12	12	12	12
Wheat bran	10	10	10	10	10
Premix ^1^	5	5	5	5	5
Nutrients					
DM, g/kg	661.3	659.1	660.3	668.6	661.5
CP, g/kg	105.1	108.1	111.8	124.0	127.5
EE, g/kg	22.0	22.5	23.8	25.1	25.9
NDF, g/kg	555.2	541.9	528.2	507.0	503.8
ADF, g/kg	279.0	257.3	247.4	240.4	240.2
Ca, g/kg	12.2	11.4	13.1	11.1	12.5
P, g/kg	8.5	8.2	8.4	8.6	8.1
ME, MJ/kg	9.1	9.2	9.5	9.7	9.9

AF = alfalfa; DM = dry matter; CP = crude protein; EE = ether extract; NDF = neutral-detergent fibre; ADF = acid-detergent fibre; ME = metabolizable energy. ^1^ The premix provided the following per kg of diets: VA 9 750 IU, VD_3_ 2 450 IU, VE 19.5 IU, nicotinic acid 11.5 mg, Fe 66 mg, Zn 70 mg, Mn 38 mg, S 0.4 mg, Se 0.3 mg, Ca 0.75 g, P ≥ 0.75 g, NaCl 9 g.

**Table 2 animals-12-02591-t002:** The effects of alfalfa–sweet sorghum (AF–SS) silage-based diets on physiological status in blood of Karakul lambs.

Items	0%AF	20%AF	40%AF	60%AF	80%AF	SEM ^1^	Linear	Quadratic
Creatinine, umol/L	86.92	93.76	95.94	96.33	90.67	1.18	0.53	0.14
BUN, mmol/L	4.56	4.79	5.10	5.25	5.96	0.59	0.24	0.62
T-AOC, U/mL	4.86 ^b^	5.59 ^b^	6.21 ^ab^	7.32 ^a^	7.60 ^a^	0.74	0.02	0.84
SOD, U/mL	97.81 ^b^	99.56 ^b^	110.33 ^a^	108.46 ^a^	101.23 ^b^	1.48	<0.01	<0.01
T-CHO, mmol/L	2.61	2.92	2.98	3.08	3.00	0.30	0.59	0.36

AF = alfalfa; BUN = blood urea nitrogen; T-AOC = total antioxidant capacity; SOD = superoxide dismutase; T-CHO = total cholesterol. ^1^ SEM = standard error of the mean; n = 6 for each group. ^a,b^ Significant differences at the level of *p* < 0.05.

**Table 3 animals-12-02591-t003:** The effects of alfalfa–sweet sorghum (AF–SS) silage-based diets on DM intake and physiological status in rumen of Karakul lambs.

Items	0%AF	20%AF	40%AF	60%AF	80%AF	SEM ^1^	Linear	Quadratic
DM intake, kg/d	0.78 ^b^	0.85 ^b^	0.86 ^b^	0.92 ^a^	0.89 ^a^	0.02	<0.01	<0.01
pH	6.79	6.74	6.69	6.54	6.46	0.11	0.09	0.60
NH_3_-N, mg/dL	13.16 ^b^	13.76 ^b^	14.87 ^a^	15.53 ^a^	14.66 ^a^	0.43	0.02	0.06
tVFA, mmol/L	104.82 ^c^	112.82 ^bc^	123.90 ^ab^	128.77 ^a^	117.73 ^ab^	5.04	<0.01	<0.01
Acetate, mmol/L	59.34 ^c^	63.68 ^bc^	71.17 ^ab^	74.28 ^a^	64.43 ^bc^	3.74	0.03	<0.01
Propionate, mmol/L	28.58	29.63	31.94	32.13	32.06	1.81	0.23	0.39
Butyrate, mmol/L	12.71	15.12	15.90	17.03	16.00	1.35	0.08	0.10
Valerate, mmol/L	1.79	1.90	2.12	2.28	2.36	0.37	0.10	0.91
Isobutyrate, mmol/L	0.92	0.97	1.10	1.22	1.16	0.10	0.07	0.43
Isovalerate, mmol/L	1.46	1.51	1.67	1.84	1.73	0.13	0.07	0.36
Acetate:Propionate	2.08	2.15	2.23	2.32	2.01	0.13	0.88	0.06

AF = alfalfa; DM = dry matter; NH_3_-N = ammonia-nitrogen; tVFA = total volatile fatty acid. ^1^ SEM standard error of the mean; n = 6 for each group. ^a,b,c^ Significant differences at the level of *p* < 0.05.

**Table 4 animals-12-02591-t004:** Analysis of alpha diversity.

Indices	0%AF	20%AF	40%AF	60%AF	80%AF	SEM ^1^	Linear	Quadratic
Observed species	1649	1624	1420	1686	1617	188.75	0.67	0.47
Shannon	6.78	7.75	7.68	7.81	8.19	0.46	0.10	0.44
Simpson	0.90	0.98	0.99	0.98	0.99	0.04	0.21	0.17
Chao1	1869.43	1891.07	1591.29	1990.49	1748.07	206.18	0.41	0.76
ace	1940.82	1946.88	1651.33	1985.36	1795.63	218.76	0.55	0.69

AF = alfalfa. ^1^ SEM standard error of the mean; n = 3 for each group.

**Table 5 animals-12-02591-t005:** Composition and relative abundance of rumen bacterial communities at phylum.

Phylum	0%AF	20%AF	40%AF	60%AF	80%AF	SEM ^1^	Linear	Quadratic
*Bacteroidetes*	59.95	56.35	42.44	50.73	52.41	5.70	0.09	0.06
*Firmicutes*	34.82 ^b^	38.86 ^b^	51.66 ^a^	41.21 ^ab^	40.93 ^ab^	4.63	0.04	0.03
*Proteobacteria*	1.20	0.85	1.19	1.65	1.20	0.72	0.86	0.97
*Synergistetes*	1.37	0.49	0.46	0.27	0.24	0.68	0.49	0.42
*Actinobacteria*	0.73 ^b^	0.81 ^b^	0.75 ^b^	2.12 ^a^	2.20 ^a^	0.45	0.01	0.26
*Spirochaetes*	0.30	0.61	0.05	1.08	0.32	0.66	0.60	0.76
*unidentified_Bacteria*	0.29 ^b^	0.31 ^b^	0.77 ^a^	0.54 ^ab^	0.36 ^b^	0.11	<0.01	<0.01
*Melainabacteria*	0.19 ^b^	0.33 ^b^	0.35 ^b^	1.12 ^a^	1.13 ^a^	0.32	0.03	0.56
*Tenericutes*	0.19	0.27	0.20	0.16	0.18	0.04	0.19	0.43
*Euryarchaeota*	0.12 ^b^	0.39 ^b^	1.54 ^a^	0.16 ^b^	0.47 ^b^	0.42	0.04	0.05
Others	0.22	0.34	0.31	0.25	0.15	0.08	0.31	0.07
Total	99.37	99.62	99.73	99.28	99.60	0.28	0.87	0.58

AF = alfalfa. ^1^ SEM standard error of the mean; n = 3 for each group. ^a,b^ Significant differences at the level of *p* < 0.05.

**Table 6 animals-12-02591-t006:** Composition and relative abundance of rumen bacterial communities at genus.

Genus	0%AF	20%AF	40%AF	60%AF	80%AF	SEM ^1^	Linear	Quadratic
*unidentified_Bacteroidales*	21.51 ^a^	16.16 ^b^	4.39 ^c^	4.32 ^c^	3.30 ^c^	9.07	0.02	0.42
*unidentified_Lachnospiraceae*	4.13	2.26	6.49	2.50	2.95	1.34	0.50	0.34
*unidentified_Ruminococcaceae*	2.56	2.97	3.33	2.61	2.87	0.44	0.80	0.26
*unidentified_Rikenellaceae*	2.20	3.38	0.77	0.59	0.68	1.44	0.30	0.97
*Fretibacterium*	1.37	0.49	0.46	0.27	0.22	0.68	0.48	0.43
*Papillibacter*	1.19	1.73	2.26	1.05	1.21	0.41	0.07	0.05
*Saccharofermentans*	0.98	0.77	1.45	1.28	0.93	0.31	0.26	0.20
*unidentified_Clostridiales*	0.74	0.98	0.63	1.59	1.43	0.41	0.06	0.65
*Succiniclasticum*	0.61 ^c^	1.11 ^b^	2.43 ^a^	1.12 ^b^	0.91 ^c^	0.56	0.63	0.02
*unidentified_Prevotellaceae*	0.55	1.08	0.47	1.45	1.09	0.44	0.18	0.88
*Romboutsia*	0.34	0.13	0.07	0.59	0.15	0.26	0.32	0.88
*Candidatus_Saccharimonas*	0.25	0.29	0.72	0.52	0.34	0.13	0.18	0.21
*Bacteroides*	0.23	0.16	0.26	0.20	0.20	0.10	0.90	0.96
*Quinella*	0.22	0.35	1.59	0.13	0.11	0.36	0.59	0.11
*Acetitomaculum*	0.21 ^b^	0.18 ^b^	0.20 ^b^	0.53 ^a^	0.65 ^a^	0.14	<0.01	0.13
*Enterorhabdus*	0.16 ^c^	0.28 ^bc^	0.17 ^c^	0.62 ^a^	0.80 ^a^	0.17	<0.01	0.17
*Methanobrevibacter*	0.11	0.37	1.51	0.15	0.45	0.42	0.64	0.06
*unidentified_Christensenellaceae*	0.11	0.08	0.08	0.13	0.12	0.03	0.43	0.41
*Fibrobacter*	0.10	0.08	0.04	0.09	0.06	0.04	0.39	0.59
*Bacillus*	0.06	0.27	0.01	0.09	0.16	0.11	0.22	0.84
Others	58.39	62.80	68.85	74.66	76.68	8.58	0.24	0.83
Total	95.82	95.93	96.19	94.48	95.31	1.34	0.73	0.89

AF = alfalfa. ^1^ SEM standard error of the mean; n = 3 for each group. ^a,b,c^ Significant differences at the level of *p* < 0.05.

**Table 7 animals-12-02591-t007:** The relative abundance of rumen bacterial species.

Species	0%AF	20%AF	40%AF	60%AF	80%AF	SEM ^1^	Linear	Quadratic
*Ruminococcus_flavefaciens*	0.33	0.31	0.47	0.22	0.27	0.10	0.40	0.30
*Ruminococcus_albus*	0.06 ^b^	0.09 ^ab^	0.09 ^ab^	0.10 ^ab^	0.19 ^a^	0.05	0.04	0.35
*Butyrivibrio_fibrisolvens*	0.45	0.39	0.49	0.44	0.46	0.36	0.93	0.12
*Clostridium_disporicum*	0.05	0.06	0.02	0.06	0.08	0.03	0.31	0.22

AF = alfalfa. ^1^ SEM standard error of the mean; n = 3 for each group. ^a,b^ Significant differences at the level of *p* < 0.05.

## Data Availability

The data presented in this study are available on request from the corresponding author.

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
