# Peer review of "Silage Mixtures of Alfalfa with Sweet Sorghum Alter Blood and Rumen Physiological Status and Rumen Microbiota of Karakul Lambs"

_animals, 2022, doi:10.3390/ani12192591_

Round 1

Reviewer 1 Report

The paper contains valuable data. Results were properly reported, and the findings have been accurately discussed and compared with other published papers. For further improvement of the manuscript, it requires some modification.

P2,L50

The introduction needs to be entirely re-written. It is very vague, and does not give the reader the necessary context to understand why you used the treatments you did, and why you made the measurements that you did. 

You can use new references such as:

BESHARATI, M., PALANGI, V., NIAZIFAR, M., & NEMATI, Z. (2020). Comparison study of flaxseed, cinnamon and lemon seed essential oils additives on quality and fermentation characteristics of lucerne silage. Acta Agriculturae Slovenica, 115(2), 455-462.

Sharifi, M., Taghizadeh, A., Hosseinkhani, A., Mohammadzadeh, H., Palangi, V., Macit, M., ... & Abachi, S. (2022). Nitrate supplementation at two forage levels in dairy cows feeding: milk production and composition, fatty acid profiles, blood metabolites, ruminal fermentation, and hydrogen sink. Anim. Sci, 22(2).

P2,L82

Please move the experimental animal ethics committee protocol no.

Table 1. P2,L93

The treatments are unclear, for example 0, 20, 40 and … AF should be shown in table clearly.

Reviewer 2 Report

The authors’ idea of mixing alfalfa and sweet sorgo for silage making can be evaluated as good, considering the specifies of the both feed sources: alfalfa is low in dry matter and sugar, but high in protein; the sorghum is high in sugar, dry matter and biomass, and has drought tolerant. In other words, the both silage raw materials are very suitable for mixing when preparing a silage, they are very different, but complement each other very well. The authors present precisely these characteristics of both feed sources in their introductory section. Sufficient references have been used.  From a scientific point of view, the effect of this type of silage on growth performance of lambs is also interesting.

The aim of the article is clearly stated. All methods used are described very carefully and with high professionalism. The obtained results are presented in six Tables and illustrated in three Figures. All of them are well presented, without any obscure details. The descriptions of all measured parameters made are clear and accurately reflect the data from the tables and/or figures.

My suggestion:

P (level of significance: p < 0.05; p > 0.05; p < 0.01; p = 0.001) be excluded from the abstract text.

The title of the article could be better as follows: Can Silage Mixtures of Alfalfa with Sweet Sorghum Alter Blood and Rumen Physiological Status and Rumen Microbiota of Karakul Lambs?

Criticism

The research design is not adequate enough. Table 1 shows that the five diets used were not well balanced in term of ME, crude protein and ether extract (higher than control), NDF and ADF (lower than control). In such a case it is not certain whether the differences obtained in the experimental groups, compared to the control, are due to the type of silage or to unbalanced diets. Please, let the authors try to discuss this uncertainty.

Noticed weaknesses:

Please, add "post mortem" regarding rumen fluid sampling (l.109).

In vitro (italic) – l.72 

Reviewer 3 Report

Authors conducted a good research

however it needs some improvement before publication i have mentioned my comments below 

Title of manuscript must be changed like effects of alfalfa-sweet sorghum mixed silages on 22 physiological state in blood and rumen, and ruminal microbiota of sheep/ or any other 

L29 and delete between blood and rumen.

L30 please mentioned the average weight and age of the lambs used in current experiment 

L31 What is 40% AF-SS mixed please clear it 

please add some more lines of Materials method section

please rewrite lines from  51-54

The introduction section is not enough you have to add more material regarding the literature of the study.

  Please add animal ethical code and university approval in Material method section.

In silage preparation did you used any inoculant? 

it is not clear due to age? that how you have provided silage to lambs 

Please mentioned that how you have prepared the diet of the lambs add the reference.

L100-101 rewrite please

L100 please briefly describe the procedure as mentioned wang et al.

regarding slaughtering, growth trait, nutrient digestibility and meat quality

 L106 from which place blood was collected 

delete the space in 3000g L107

L-107-108 rewrite please 

L 108-113 Please rewrite

2.3 subtitle must be changed

please add a separate section for Rumen content determination  

L-115-118 add brief procedure and reference please

add brief procedure of the tVFA determination method.

add the name of the researcher reference no (14) L 130

i suggest rewrite the 2.4 subtitle and add brief procedure and related references 

In result section please follow the same sequence in results which are mentioned in tables.

L 155-156 rewrite

please add significant difference word in results section in all results where you get difference 

 The language used in result section is not appropriate please Rewrite the results section

Please add some more references (2018-2022) and validate the results of the current study with previous findings.

conclusion please revise the conclusion section 

Reviewer 4 Report

Dear authors,

The manuscript titled 'Can silage mixtures of alfalfa with sweet sorghum modify physiological state in blood and rumen, and ruminal microbiota of Karabul lambs?' is addressing an interesting topic of research for publication at Animals MDPI. See below a few comments to improve its quality:

L48-49 Replace 'Kar-akul' by 'Ka-rakul'.

L84 Use scientific nomenclature to cite both varieties.

L88 Write the brand and model of the silage machine used.

L89 Explain how silage mixtures were stored.

L90  Write the brand and model of the rations mixer used.

L106 Explain how blood samples were collected.

L125 Replace 'Bioinformation' by 'Bioinformatic'.

L163 Replace 'of means' by 'of the mean'.

L181 Replace 'of means' by 'of the mean'.

L188 Replace 'of means' by 'of the mean'.

L190 Replace 'There' by 'there'.

L199 Replace 'founded' by 'found'.

L215 Replace 'of means' by 'of the mean'.

L218 Replace 'founded' by 'found'.

L219 Replace '3.3' by '3.30'.

L237 Replace 'of means' by 'of the mean'.

L245 Replace 'of means' by 'of the mean'.

L336 Replace 'to more search in future' by 'to search more in the future'.

L338 Replace 'confirmed' by 'showed'.

Best regards,

Reviewer.

Round 2

Reviewer 3 Report

Please revise title "Silage Mixtures of Alfalfa with Sweet Sorghum Alter Blood and Rumen Physiological Status and Rumen Microbiota of Karakul Lambs"

As a recommended that please add more material in introduction section regarding your title but you have not added the major issues and solutions.

In material method section please add animal ethical code and guidelines.

What about dry matter intake ? 

please add this data  because this data is important when you are studying rumen fermentation and Microbiota.

 Add references of  the brief method you have mentioned in Material Method Section .

Results section revised very well 

Conclusion section revised well 
